# Volley Revolver: A Novel Matrix-Encoding Method for Privacy-Preserving Neural Networks (Inference)

## Abstract

In this work, we present a novel matrix-encoding method that is particularly convenient for neural networks to make predictions in a privacy-preserving manner using homomorphic encryption. Based on this encoding method, we implement a convolutional neural network for handwritten image classification over encryption. For two matrices $A$ and $B$ to perform homomorphic multiplication, the main idea behind it, in a simple version, is to encrypt matrix $A$ and the transpose of matrix $B$ into two ciphertexts respectively. With additional operations, the homomorphic matrix multiplication can be calculated over encrypted matrices efficiently. For the convolution operation, we in advance span each convolution kernel to a matrix space of the same size as the input image so as to generate several ciphertexts, each of which is later used together with the ciphertext encrypting input images for calculating some of the final convolution results. We accumulate all these intermediate results and thus complete the convolution operation.

In a public cloud with 40 vCPUs, our convolutional neural network implementation on the MNIST testing dataset takes $\sim 287$ seconds to compute ten likelihoods of 32 encrypted images of size $28 \times 28$ simultaneously. The data owner only needs to upload one ciphertext ($\sim 19.8$ MB) encrypting these 32 images to the public cloud.

## 1 Introduction

Machine learning applied in some specific domains such as health and finance should preserve privacy while processing private or confidential data to make accurate predictions. In this study, we focus on privacy-preserving neural network inference, which aims to outsource a well-trained inference model to a cloud service in order to make predictions on private data. For this purpose, the data should be encrypted first and then sent to the cloud service that should not be capable of having access to the raw data. Compared to other cryptology technologies such as Secure Multi-Party Computation, Homomorphic Encryption (HE) provides the most stringent security for this task.

Combining HE with Convolutional Neural Networks (CNN) inference has been receiving more and more attention in recent years since Gilad-Bachrach et al. [6] proposed a framework called `Cryptonets`. `Cryptonets` applies neural networks to make accurate inferences on encrypted data with high throughput. Chanranne et al. [2] extended this work to deeper CNN using a different underlying software library called `HElib` [7] and leveraged batch normalization and training process to develop better quality polynomial approximations of the `ReLU` function for stability and accuracy. Chou et al. [4] developed a pruning and quantization approach with other deep-learning optimization techniques and presented a method for encrypted neural networks inference, `Faster CryptoNets`. Brutzkus et al. [1] developed new encoding methods other than the one used in `Cryptonets` for representing data and presented the Low-Latency CryptoNets (`LoLa`) solution. Jiang et al. [9]

proposed an efficient evaluation strategy for secure outsourced matrix multiplication with the help of a novel matrix-encoding method.

**Contributions** In this study, our contributions are in three main parts:

1. We introduce a novel data-encoding method for matrix multiplications on encrypted matrices, `Volley Revolver`, which can be used to multiply matrices of arbitrary shape efficiently.

2. We propose a feasible evaluation strategy for convolution operation, by devising an efficient homomorphic algorithm to sum some intermediate results of convolution operations.

3. We develop some simulated operations on the packed ciphertext encrypting an image dataset as if there were multiple virtual ciphertexts inhabiting it, which provides a compelling new perspective of viewing the dataset as a three-dimensional structure.

## 2  Preliminaries

Let "$\oplus$" and "$\otimes$" denote the component-wise addition and multiplication respectively between ciphertexts encrypting matrices and the ciphertext $\texttt{ct}.P$ the encryption of a matrix $P$. Let $I_{[i][j]}^{(m)}$ represent the single pixel of the $j$-th element in the $i$-th row of the $m$-th image from the dataset.

**Homomorphic Encryption**   Homomorphic Encryption is one kind of encryption but has its characteristic in that over an HE system operations on encrypted data generate ciphertexts encrypting the right results of corresponding operations on plaintext without decrypting the data nor requiring access to the secret key. Since Gentry [5] presented the first fully homomorphic encryption scheme, tackling the over three decades problem, much progress has been made on an efficient data encoding scheme for the application of machine learning to HE. Cheon et al. [3] constructed an HE scheme (CKKS) that can deal with this technique problem efficiently, coming up with a new procedure called `rescaling` for approximate arithmetic in order to manage the magnitude of plaintext. Their open-source library, `HEAAN`, like other HE libraries also supports the Single Instruction Multiple Data (aka SIMD) manner [11] to encrypt multiple values into a single ciphertext.

Given the security parameter, `HEAAN` outputs a secret key $sk$, a public key $pk$, and other public keys used for operations such as rotation. For simplicity, we will ignore the `rescale` operation and deem the following operations to deal with the magnitude of plaintext automatedly. `HEAAN` has the following functions to support the HE scheme:

1. $\texttt{Enc}_{pk}(m)$: For the public key $pk$ and a message vector $m$, `HEAAN` encrypts the message $m$ into a ciphertext $\texttt{ct}$.

2. $\texttt{Dec}_{sk}(\texttt{ct})$: Using the secret key, this algorithm returns the message vector encrypted by the ciphertext $\texttt{ct}$.

3. $\texttt{Add}(\texttt{ct}_1, \texttt{ct}_2)$: This operation returns a new ciphertext that encrypts the message $\text{Dec}_{sk}(\texttt{ct}_1) \oplus \text{Dec}_{sk}(\texttt{ct}_2)$.

4. $\texttt{Mul}(\texttt{ct}_1, \texttt{ct}_2)$: This procedure returns a new ciphertext that encrypts the message $\text{Dec}_{sk}(\texttt{ct}_1) \otimes \text{Dec}_{sk}(\texttt{ct}_2)$.

5. $\texttt{Rot}(\texttt{ct}, l)$: This procedure generates a ciphertext encrypting a new plaintext vector obtained by rotating the the original message vector $m$ encrypted by $\texttt{ct}$ to the left by $l$ positions.

**Database Encoding Method**   For brevity, we assume that the training dataset has $n$ samples with $f$ features and that the number of slots in a single ciphertext is at least $n \times f$. A training dataset is usually organized into a matrix $Z$ each row of which represents an example. Kim et al. [10] propose an efficient database encoding method to encrypt this matrix into a single ciphertext in a row-by-row manner. They provide two basic but important shifting operations by shifting $1$ and $f$ positions respectively: the *incomplete* column shifting and the row shifting. The matrix obtained from matrix $Z$ by the *incomplete* column shifting operation is shown as follows:

$$Z = \begin{bmatrix} z_{[1][1]} & z_{[1][2]} & \cdots & z_{[1][f]} \\ z_{[2][1]} & z_{[2][2]} & \cdots & z_{[2][f]} \\ \vdots & \vdots & \ddots & \vdots \\ z_{[n][1]} & z_{[n][2]} & \cdots & z_{[n][f]} \end{bmatrix} \xmapsto{\textit{incomplete} \text{ column shifting}} \begin{bmatrix} z_{[1][2]} & z_{[1][3]} & \cdots & z_{[2][1]} \\ z_{[2][2]} & z_{[2][3]} & \cdots & z_{[3][1]} \\ \vdots & \vdots & \ddots & \vdots \\ z_{[n][2]} & z_{[n][3]} & \cdots & z_{[1][1]} \end{bmatrix}.$$

Han et al. [8] summarize another two procedures, `SumRowVec` and `SumColVec`, to compute the summation of each row and column respectively. The results of two procedures on $Z$ are as follows:

$$\texttt{SumRowVec}(Z) = \begin{bmatrix} \sum_{i=1}^{n} z_{[i][1]} & \sum_{i=1}^{n} z_{[i][2]} & \cdots & \sum_{i=1}^{n} z_{[i][f]} \\ \sum_{i=1}^{n} z_{[i][1]} & \sum_{i=1}^{n} z_{[i][2]} & \cdots & \sum_{i=1}^{n} z_{[i][f]} \\ \vdots & \vdots & \ddots & \vdots \\ \sum_{i=1}^{n} z_{[i][1]} & \sum_{i=1}^{n} z_{[i][2]} & \cdots & \sum_{i=1}^{n} z_{[i][f]} \end{bmatrix},$$

$$\texttt{SumColVec}(Z) = \begin{bmatrix} \sum_{j=1}^{f} z_{[1][j]} & \sum_{j=1}^{f} z_{[1][j]} & \cdots & \sum_{j=1}^{f} z_{[1][j]} \\ \sum_{j=1}^{f} z_{[2][j]} & \sum_{j=1}^{f} z_{[2][j]} & \cdots & \sum_{j=1}^{f} z_{[2][j]} \\ \vdots & \vdots & \ddots & \vdots \\ \sum_{j=1}^{f} z_{[n][j]} & \sum_{j=1}^{f} z_{[n][j]} & \cdots & \sum_{j=1}^{f} z_{[n][j]} \end{bmatrix}.$$

We propose a new useful procedure called `SumForConv` to facilitate convolution operation for every image. Below we illustrate the result of `SumForConv` on $Z$ taking the example that $n$ and $f$ are both 4 and the kernel size is $3 \times 3$:

$$Z = \begin{bmatrix} z_{[1][1]} & z_{[1][2]} & z_{[1][3]} & z_{[1][4]} \\ z_{[2][1]} & z_{[2][2]} & z_{[2][3]} & z_{[2][4]} \\ z_{[3][1]} & z_{[3][2]} & z_{[3][3]} & z_{[3][4]} \\ z_{[4][1]} & z_{[4][2]} & z_{[4][3]} & z_{[4][4]} \end{bmatrix} \xmapsto{\texttt{SumForConv}(\cdot,3,3)} \begin{bmatrix} s_{[1][1]} & s_{[1][2]} & 0 & 0 \\ s_{[2][1]} & s_{[2][2]} & 0 & 0 \\ 0 & 0 & 0 & 0 \\ 0 & 0 & 0 & 0 \end{bmatrix},$$

where $s_{[i][j]} = \sum_{p=i}^{i+2} \sum_{q=j}^{j+2} z_{[p][q]}$ for $1 \le i, j \le 2$. In the convolutional layer, `SumForConv` can help to compute some partial results of convolution operation for an image simultaneously.

## 3  Technical details

We introduce a novel matrix-encoding method called `Volley Revolver`, which is particularly suitable for secure matrix multiplication. The basic idea is to place each semantically-complete information (such as an example in a dataset) into the corresponding row of a matrix and encrypt this matrix into a single ciphertext. When applying it to private neural networks, `Volley Revolver` puts the whole weights of every neural node into the corresponding row of a matrix, organizes all the nodes from the same layer into this matrix, and encrypts this matrix into a single ciphertext.

### 3.1  Encoding Method for Matrix Multiplication

Suppose that we are given an $m \times n$ matrix $A$ and a $n \times p$ matrix $B$ and suppose to compute the matrix $C$ of size $m \times p$, which is the matrix product $A \cdot B$ with the element $C_{[i][j]} = \sum_{k=1}^{n} a_{[i][k]} \times b_{[k][j]}$:

$$A = \begin{bmatrix} a_{[1][1]} & a_{[1][2]} & \cdots & a_{[1][n]} \\ a_{[2][1]} & a_{[2][2]} & \cdots & a_{[2][n]} \\ \vdots & \vdots & \ddots & \vdots \\ a_{[m][1]} & a_{[n][2]} & \cdots & a_{[m][n]} \end{bmatrix}, B = \begin{bmatrix} b_{[1][1]} & b_{[1][2]} & \cdots & b_{[1][p]} \\ b_{[2][1]} & b_{[2][2]} & \cdots & b_{[2][p]} \\ \vdots & \vdots & \ddots & \vdots \\ b_{[n][1]} & b_{[n][2]} & \cdots & b_{[n][p]} \end{bmatrix}.$$

For simplicity, we assume that each of the three matrices $A$, $B$ and $C$ could be encrypted into a single ciphertext. We also make the assumption that $m$ is greater than $p$, $m > p$. We will not illustrate the other cases where $m \le p$, which is similar to this one. When it comes to the homomorphic matrix multiplication, `Volley Revolver` encodes matrix $A$ directly but encodes the padding form of the transpose of matrix $B$, by using two row-ordering encoding maps. For matrix $A$, we adopt the same encoding method that [9] did by the encoding map $\tau_a : A \mapsto \bar{A} = (a_{[1+(k/n)][1+(k\%n)]})_{0 \le k < m \times n}$. For matrix $B$, we design a very different encoding method from [9] for `Volley Revolver` : we transpose the matrix $B$ first and then extend the resulting matrix in

the vertical direction to the size $m \times n$. Therefore `Volley Revolver` adopts the encoding map $\tau_b : B \mapsto \bar{B} = (b_{[1+(k\%n)][1+((k/n)\%p)]})_{0 \le k < m \times n}$, obtaining the matrix from mapping $\tau_b$ on $B$:

$$
\begin{bmatrix}
b_{[1][1]} & b_{[1][2]} & \cdots & b_{[1][p]} \\
b_{[2][1]} & b_{[2][2]} & \cdots & b_{[2][p]} \\
\vdots & \vdots & \ddots & \vdots \\
b_{[n][1]} & b_{[n][2]} & \cdots & b_{[n][p]}
\end{bmatrix}
\xmapsto{\tau_b}
\begin{bmatrix}
b_{[1][1]} & b_{[2][1]} & \cdots & b_{[n][1]} \\
b_{[1][2]} & b_{[2][2]} & \cdots & b_{[n][2]} \\
\vdots & \vdots & \ddots & \vdots \\
b_{[1][p]} & b_{[2][p]} & \cdots & b_{[n][p]} \\
b_{[1][1]} & b_{[2][1]} & \cdots & b_{[n][1]} \\
\vdots & \vdots & \ddots & \vdots \\
b_{[1][1+(m-1)\%p]} & b_{[2][1+(m-1)\%p]} & \cdots & b_{[n][1+(m-1)\%p]}
\end{bmatrix}.
$$

**Homomorphic Matrix Multiplication** We report an efficient evaluation algorithm for homomorphic matrix multiplication. This algorithm uses a ciphertext `ct.R` encrypting zeros or a given value such as the weight bias of a fully-connected layer as an accumulator and an operation `RowShifter` to perform a specific kind of row shifting on the encrypted matrix $\bar{B}$. `RowShifter` pops up the first row of $\bar{B}$ and appends another corresponding already existing row of $\bar{B}$:

$$
\begin{bmatrix}
b_{[1][1]} & b_{[2][1]} & \cdots & b_{[n][1]} \\
b_{[1][2]} & b_{[2][2]} & \cdots & b_{[n][2]} \\
\vdots & \vdots & \ddots & \vdots \\
b_{[1][p]} & b_{[2][p]} & \cdots & b_{[n][p]} \\
b_{[1][1]} & b_{[2][1]} & \cdots & b_{[n][1]} \\
\vdots & \vdots & \ddots & \vdots \\
b_{[1][r]} & b_{[2][r]} & \cdots & b_{[n][r]}
\end{bmatrix}
\xmapsto{\text{RowShifter}(\bar{B})}
\begin{bmatrix}
b_{[1][2]} & b_{[2][2]} & \cdots & b_{[n][2]} \\
\vdots & \vdots & \ddots & \vdots \\
b_{[1][p]} & b_{[2][p]} & \cdots & b_{[n][p]} \\
b_{[1][1]} & b_{[2][1]} & \cdots & b_{[n][1]} \\
\vdots & \vdots & \ddots & \vdots \\
b_{[1][r]} & b_{[2][r]} & \cdots & b_{[n][r]} \\
b_{[1][(r+1)\%p]} & b_{[2][(r+1)\%p]} & \cdots & b_{[n][(r+1)\%p]}
\end{bmatrix}.
$$

For two ciphertexts `ct.A` and `ct.B̄`, the algorithm for homomorphic matrix multiplication has $p$ iterations. For the $k$-th iteration where $0 \le k < p$ there are the following four steps:

*Step 1.* This step uses `RowShifter` on `ct.B̄` to generate a new ciphertext `ct.B̄₁` and then computes the homomorphic multiplication between ciphertexts `ct.A` and `ct.B̄₁` to get the resulting product `ct.AB̄₁`. When $k = 0$, in this case `RowShifter` just return a copy of the ciphertext `ct.B̄`.

*Step 2.* In this step, the public cloud applies `SumColVec` on `ct.AB̄₁` to collect the summation of the data in each row of $A\bar{B}_1$ for some intermediate results, and obtain the ciphertext `ct.D`.

*Step 3.* This step designs a special matrix $F$ to generate a ciphertext `ct.F` for filtering out the redundancy element in $D$ by one multiplication $\text{Mul}(\text{ct.}F, \text{ct.}D)$, resulting the ciphertext `ct.D₁`.

*Step 4.* The ciphertext `ct.R` is then used to accumulate the intermediate ciphertext `ct.D₁`.

The algorithm will repeat Steps 1 to 4 for $p$ times and finally aggregates all the intermediate ciphertexts, returning the ciphertext `ct.C`. Algorithm 1 shows how to perform our homomorphic matrix multiplication. Figure 1 describes a simple case for Algorithm 1 where $m = 2$, $n = 4$ and $p = 2$.

The calculation process of this method, especially for the simple case where $m = p$, is intuitively similar to a special kind of revolver that can fire multiple bullets at once (The first matrix $A$ is settled still while the second matrix $B$ is revolved). That is why we term our encoding method "`Volley Revolver`". In the real-world cases where $m \mod p = 0$, the operation `RowShifter` can be reduced to only need one rotation `RowShifter = Rot(ct, n)`, which is much more efficient and should thus be adopted whenever possible. Corresponding to the neural networks, we can set the number of neural nodes for each fully-connected layer to be a power of two to achieve this goal.

### 3.2 Homomorphic Convolution Operation

In this subsection, we first introduce a novel but impractical algorithm to calculate the convolution operation for a single grayscale image of size $h \times w$ based on the assumption that this single image can *happen* to be encrypted into a single ciphertext without vacant slots left, meaning the number $N$ of slots in a packed ciphertext chance to be $N = h \times w$. We then illustrate how to use this method to compute the convolution operation of several images of *any size* at the same time for a convolutional layer after these images have been encrypted into a ciphertext and been viewed as several virtual

---

**Algorithm 1** Homomorphic matrix multiplication

---

**Input:** $\texttt{ct}.A$ and $\texttt{ct}.\bar{B}$ for $A \in \mathbb{R}^{m \times n}$, $B \in \mathbb{R}^{n \times p}$ and $B \xmapsto{\texttt{Volley Revolver Encoding}} \bar{B} \in \mathbb{R}^{m \times n}$
**Output:** The encrypted resulting matrixs $\texttt{ct}.C$ for $C \in \mathbb{R}^{m \times p}$ of the matrix product $A \cdot B$

 1: Set $C \leftarrow \mathbf{0}$               ▷ $C$: To accumulate intermediate matrices
 2: $\texttt{ct}.C \leftarrow \texttt{Enc}_{pk}(C)$
   ▷ The outer loop (could be computed in parallel)
 3: **for** $idx := 0$ to $p - 1$ **do**
 4:    $\texttt{ct}.T \leftarrow \texttt{RowShifter}(\texttt{ct}.\bar{B}, p, idx)$
 5:    $\texttt{ct}.T \leftarrow \texttt{Mul}(\texttt{ct}.A, \texttt{ct}.T)$
 6:    $\texttt{ct}.T \leftarrow \texttt{SumColVec}(\texttt{ct}.T)$
    ▷ Build a specifically-designed matrix to clean up the redundant values
 7:    Set $F \leftarrow \mathbf{0}$              ▷ $F \in \mathbb{R}^{m \times n}$
 8:    **for** $i := 1$ to $m$ **do**
 9:     $F[i][(i + idx)\%n] \leftarrow 1$
10:    **end for**
11:    $\texttt{ct}.T \leftarrow \texttt{Mul}(\texttt{Enc}_{pk}(F), \texttt{ct}.T)$
    ▷ To accumulate the intermediate results
12:    $\texttt{ct}.C \leftarrow \texttt{Add}(\texttt{ct}.C, \texttt{ct}.T)$
13: **end for**
14: **return** $\texttt{ct}.C$

---

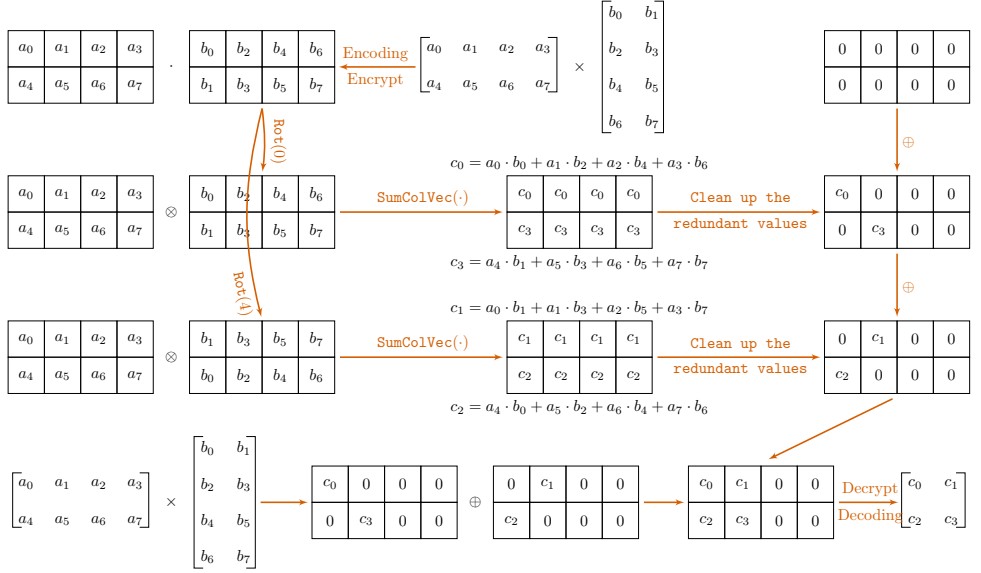

Figure 1: Our matrix multiplication algorithm with $m = 2$, $n = 4$ and $p = 2$

ciphertexts inhabiting this real ciphertext. For simplicity, we assume that the image is grayscale and that the image dataset can be encrypted into a single ciphertext.

**An impractical algorithm** Given a grayscale image $I$ of size $h \times w$ and a kernel $K$ of size $k \times k$ with its bias $k_0$ such that $h$ and $w$ are both greater than $k$, based on the assumption that this image can *happen* to be encrypted into a ciphertext $\texttt{ct}.I$ with no more or less vacant slots, we present an efficient algorithm to compute the convolution operation. We set the stride size to the usual default value $(1, 1)$ and adopt no padding technique in this algorithm.

Before the algorithm starts, the kernel $K$ should be called by an operation that we term $\texttt{Kernelspanner}$ to in advance generate $k^2$ ciphertexts for most cases where $h \geq 2 \cdot k - 1$ and $w \geq 2 \cdot k - 1$, each of which encrypts a matrix $P_i$ for $1 \leq i \leq k^2$, using a map to span the $k \times k$ kernel to a $h \times w$ matrix space. For a simple example that $h = 4$, $w = 4$ and $k = 2$, $\texttt{Kernelspanner}$

generates 4 ciphertexts and the kernel bias $k_0$ will be used to generate a ciphertext:

$$\begin{bmatrix} k_1 & k_2 \\ k_3 & k_4 \end{bmatrix} \xmapsto[\mathbb{R}^{k \times k} \mapsto k^2 \cdot \mathbb{R}^{h \times w}]{\texttt{Kernelspanner}} \begin{bmatrix} k_1 & k_2 & k_1 & k_2 \\ k_3 & k_4 & k_3 & k_4 \\ k_1 & k_2 & k_1 & k_2 \\ k_3 & k_4 & k_3 & k_4 \end{bmatrix}, \begin{bmatrix} 0 & k_1 & k_2 & 0 \\ 0 & k_3 & k_4 & 0 \\ 0 & k_1 & k_2 & 0 \\ 0 & k_3 & k_4 & 0 \end{bmatrix}, \begin{bmatrix} 0 & 0 & 0 & 0 \\ k_1 & k_2 & k_1 & k_2 \\ k_3 & k_4 & k_3 & k_4 \\ 0 & 0 & 0 & 0 \end{bmatrix},$$

$$[k_0] \mapsto Enc \begin{bmatrix} k_0 & k_0 & k_0 & 0 \\ k_0 & k_0 & k_0 & 0 \\ k_0 & k_0 & k_0 & 0 \\ 0 & 0 & 0 & 0 \end{bmatrix}. \qquad\qquad \begin{bmatrix} 0 & 0 & 0 & 0 \\ 0 & k_1 & k_2 & 0 \\ 0 & k_3 & k_4 & 0 \\ 0 & 0 & 0 & 0 \end{bmatrix}.$$

Our impractical homomorphic algorithm for convolution operation also needs a ciphertext $\texttt{ct}.R$ to accumulate the intermediate ciphertexts, which should be initially encrypted by the kernel bias $k_0$. This algorithm requires $k \times k$ iterations and the $i$-th iteration consists of the following four steps for $1 \le i \le k^2$:

*Step 1.* For ciphertexts $\texttt{ct}.I$ and $\texttt{ct}.P_i$, this step computes their multiplication and returns the ciphertext $\texttt{ct}.IP_i = \texttt{Mul}(\texttt{ct}.I, \texttt{ct}.P_i)$.

*Step 2.* To aggregate the values of some blocks of size $k \times k$, this step applies the procedure $\texttt{SumForConv}$ on the ciphertext $\texttt{ct}.IP_i$, obtaining the ciphertext $\texttt{ct}.D$.

*Step 3.* The public cloud generates a ciphertext encrypting a specially-designed matrix in order to filter out the garbage data in $\texttt{ct}.D$ by one multiplication, obtaining a ciphertext $\texttt{ct}.\bar{D}$.

*Step 4.* In this step, the homomorphic convolution-operation algorithm updates the accumulator ciphertext $\texttt{ct}.R$ by homomorphically adding $\texttt{ct}.\bar{D}$ to it, namely $\texttt{ct}.R = \texttt{Add}(\texttt{ct}.R, \texttt{ct}.\bar{D})$.

Note that Steps 1–3 in this algorithm can be computed in parallel with $k \times k$ threads. We describe how to compute homomorphic convolution operation in Algorithm 2 in detail. Figure 2 describes a simple case for the algorithm where $h = 3$, $w = 4$ and $k = 3$.

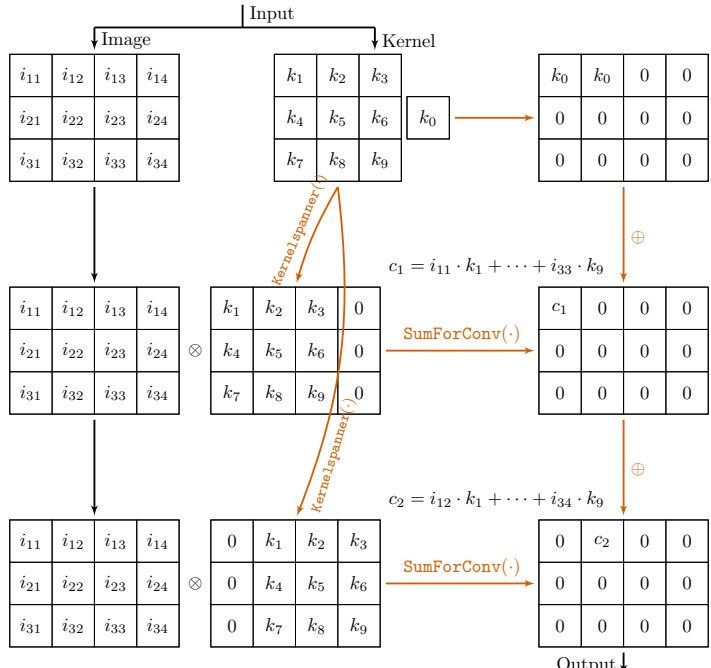

Figure 2: Our convolution operation algorithm with $h = 3$, $w = 4$ and $k = 3$

Next, we will show how to make this impractical homomorphic algorithm work efficiently in real-world cases.

---
**Algorithm 2** Homomorphic convolution operation
---
**Input:** An encrypted Image $ct.I$ for $I \in \mathbb{R}^{h \times w}$ and a kernel $K$ of size $k \times k$ with its bias $k_0$
**Output:** The encrypted resulting image $ct.I_s$ where $I_s$ has the same size as $I$
       $\triangleright$ The Third Party performs `Kernelspanner` and prepares the ciphertext encrypting kernel bias
  1: $ct.S_{[i]} \leftarrow \text{Kernelspanner}(K, h, w)$                                   $\triangleright$ $1 \leq i \leq k^2$
  2: Set $I_s \leftarrow \mathbf{0}$                                               $\triangleright$ $I_s \in \mathbb{R}^{h \times w}$
  3: **for** $i := 1$ to $h - k + 1$ **do**
  4:     **for** $j := 1$ to $w - k + 1$ **do**
  5:         $I_s[i][j] \leftarrow k_0$
  6:     **end for**
  7: **end for**
  8: $ct.I_s \leftarrow \text{Enc}_{pk}(I_s)$
      $\triangleright$ So begins the Cloud its work
  9: **for** $i := 0$ to $k - 1$ **do**
10:     **for** $j := 0$ to $k - 1$ **do**
11:         $ct.T \leftarrow \text{Mul}(ct.I, ct.S_{[i \times k + j + 1]})$
12:         $ct.T \leftarrow \text{SumForConv}(ct.T)$
           $\triangleright$ Design a matrix to filter out the redundant values
13:         Set $F \leftarrow \mathbf{0}$                                   $\triangleright$ $F \in \mathbb{R}^{m \times n}$
14:         **for** $hth := 0$ to $h - 1$ **do**
15:             **for** $wth := 0$ to $w - 1$ **do**
16:                 **if** $(wth - i) \bmod k = 0$ **and** $wth + k \leq w$ **and**
17:                     $(hth - j) \bmod k = 0$ **and** $hth + k \leq h$ **then**
18:                     $F[hth][wth] \leftarrow 1$
19:                 **end if**
20:             **end for**
21:         **end for**
22:         $ct.T \leftarrow \text{Mul}(\text{Enc}_{pk}(F), ct.T)$
           $\triangleright$ To accumulate the intermediate results
23:         $ct.I_s \leftarrow \text{Add}(ct.I_s, ct.T)$
24:     **end for**
25: **end for**
26: **return** $ct.I_s$

---

**Encoding Method for Convolution Operation** For simplicity, we assume that the dataset $X \in \mathbb{R}^{m \times f}$ can be encrypted into a single ciphertext $ct.X$, $m$ is a power of two, all the images are grayscale and have the size $h \times w$. `Volley Revolver` encodes the dataset as a matrix using the database encoding method [10] and deals with any CNN layer with a single formation. In most cases, $h \times w < f$, if this happened, zero columns could be used for padding. `Volley Revolver` extends this database encoding method [10] with some additional operations to view the dataset matrix $X$ as a three-dimensional structure.

Algorithm 2 is a feasible and efficient way to calculate the secure convolution operation in an HE domain. However, its working-environment assumption that the size of an image is exactly the length of the plaintext, which rarely happens, is too strict to make it a practical algorithm, leaving this algorithm directly useless. In addition, Algorithm 2 can only deal with one image at a time due to the assumption that a single ciphertext only encrypts only one image, which is too inefficient for real-world applications.

To solve these problems, `Volley Revolver` performs some simulated operations on the ciphertext $ct.X$ to treat the two-dimensional dataset as a three-dimensional structure. These simulated operations together could simulate the first continual space of the same size as an image of each row of the matrix encrypted in a real ciphertext as a virtual ciphertext that can perform all the HE operations. Moreover, the number of plaintext slots is usually set to a large number and hence a single ciphertext could encrypt several images. For example, the ciphertext encrypting the dataset $X \in \mathbb{R}^{m \times f}$ could

be used to simulate $m$ virtual ciphertexts $\mathtt{vct}_i$ for $1 \leq i \leq m$, as shown below:

$$Enc \begin{bmatrix} I^{(1)}_{[1][1]} & I^{(1)}_{[1][2]} & \cdots & I^{(1)}_{[h][w]} & 0 & \cdots & 0 \\ I^{(2)}_{[1][1]} & I^{(2)}_{[1][2]} & \cdots & I^{(2)}_{[h][w]} & 0 & \cdots & 0 \\ \vdots & \vdots & \ddots & \vdots & \vdots & \ddots & \vdots \\ I^{(m)}_{[1][1]} & I^{(m)}_{[1][2]} & \cdots & I^{(m)}_{[h][w]} & 0 & \cdots & 0 \end{bmatrix} \longrightarrow \begin{bmatrix} \mathtt{vEnc} \begin{bmatrix} I^{(1)}_{[1][1]} & \cdots & I^{(1)}_{[1][w]} \\ \vdots & \ddots & \vdots \\ I^{(1)}_{[h][1]} & \cdots & I^{(1)}_{[h][w]} \end{bmatrix} & 0 & \cdots & 0 \\ \vdots & \vdots & \ddots & \vdots \\ \mathtt{vEnc} \begin{bmatrix} I^{(m)}_{[1][1]} & \cdots & I^{(m)}_{[1][w]} \\ \vdots & \ddots & \vdots \\ I^{(m)}_{[h][1]} & \cdots & I^{(m)}_{[h][w]} \end{bmatrix} & 0 & \cdots & 0 \end{bmatrix}.$$

Similar to an HE ciphertext, a virtual ciphertext has virtual HE operations: $\mathtt{vEnc}$, $\mathtt{vDec}$, $\mathtt{vAdd}$, $\mathtt{vMul}$, $\mathtt{vRescale}$, $\mathtt{vBootstrapping}$ and $\mathtt{vRot}$. Except for $\mathtt{vRot}$, others can be all inherited from the corresponding HE operations. The HE operations, $\mathtt{Add}$, $\mathtt{Mul}$, $\mathtt{Rescale}$ and $\mathtt{Bootstrapping}$, result in the same corresponding virtual operations: $\mathtt{vAdd}$, $\mathtt{vMul}$, $\mathtt{vRescale}$ and $\mathtt{vBootstrapping}$. The virtual rotation operation $\mathtt{vRot}$ is much different from other virtual operations: it needs two rotation operations over the real ciphertext. We only need to simulate the rotation operation on these virtual ciphertexts to complete the simulation. The virtual rotation operation $\mathtt{vRot}(\mathtt{ct}, r)$, to rotate all the virtual ciphertexts dwelling in the real ciphertext $\mathtt{ct}$ to the left by $r$ positions, has the following simulation result:

$$Enc \begin{bmatrix} \mathtt{vEnc} \begin{bmatrix} I^{(1)}_{[1][1]} & \cdots & I^{(1)}_{[r/w][r\%w]} & I^{(1)}_{[(r+1)/w][(r+1)\%w]} & \cdots & I^{(1)}_{[h][w]} \end{bmatrix} & 0 & \cdots & 0 \\ \vdots & \vdots & \ddots & \vdots \\ \mathtt{vEnc} \begin{bmatrix} I^{(m)}_{[1][1]} & \cdots & I^{(m)}_{[r/w][r\%w]} & I^{(m)}_{[(r+1)/w][(r+1)\%w]} & \cdots & I^{(m)}_{[h][w]} \end{bmatrix} & 0 & \cdots & 0 \end{bmatrix}$$

$$\downarrow \mathtt{vRot}(\mathtt{ct}, r)$$

$$Enc \begin{bmatrix} \mathtt{vEnc} \begin{bmatrix} I^{(1)}_{[(r+1)/w][(r+1)\%w]} & \cdots & I^{(1)}_{[h][w]} & I^{(1)}_{[1][1]} & \cdots & I^{(1)}_{[r/w][r\%w]} \end{bmatrix} & 0 & \cdots & 0 \\ \vdots & \vdots & \ddots & \vdots \\ \mathtt{vEnc} \begin{bmatrix} I^{(m)}_{[(r+1)/w][(r+1)\%w]} & \cdots & I^{(m)}_{[h][w]} & I^{(1)}_{[1][1]} & \cdots & I^{(m)}_{[r/w][r\%w]} \end{bmatrix} & 0 & \cdots & 0 \end{bmatrix}.$$

To bring all the pieces together, we can use Algorithm 2 to perform convolution operations for several images in parallel based on the simulation virtual ciphertexts. The most efficient part of these simulated operations is that a sequence of operations on a real ciphertext results in the same corresponding operations on the multiple virtual ciphertexts, which would suffice the real-world applications.

# 4 Privacy-preserving CNN Inference

**Limitations on applying CNN to HE**    Homomorphic Encryption cannot directly compute functions such as the $\mathtt{ReLU}$ activation function. We use $\mathtt{Octave}$ to generate a degree-three polynomial by the least square method and just initialize all the activation layers with this polynomial, leaving the training process to determine the coefficients of polynomials for every activation layer. Other computation operations, such as matrix multiplication in the fully-connected layer and convolution operation in the convolutional layer, can also be performed by the algorithms we proposed above.

**Neural Networks Architecture**    We adopt the same CNN architecture as [9] but with some different hyperparameters. Our encoding method $\mathtt{Volley\ Revolver}$ can be used to build convolutional neural networks as deep as it needs. However, in this case, the computation time will therefore increase and bootstrapping will have to be used to refresh the ciphertext, resulting in more time-consuming. Table 1 gives a description of our neural networks architecture on the MNIST dataset.

# 5 Experimental Results

We use C++ to implement our homomorphic CNN inference. Our complete source code is publicly available at $\mathtt{https://anonymous.4open.science/r/HE\text{-}CNNinfer\text{-}ECA4/}$ .

Table 1: Description of our CNN on the MNIST dataset

| Layer | Description |
|---|---|
| CONV | 32 input images of size $28 \times 28$, 4 kernels of size $3 \times 3$, stride size of $(1, 1)$ |
| ACT-1 | $x \mapsto -0.00015120704 + 0.4610149 \cdot x + 2.0225089 \cdot x^2 - 1.4511951 \cdot x^3$ |
| FC-1 | Fully connecting with $26 \times 26 \times 4 = 2704$ inputs and 64 outputs |
| ACT-2 | $x \mapsto -1.5650465 - 0.9943767 \cdot x + 1.6794522 \cdot x^2 + 0.5350255 \cdot x^3$ |
| FC-2 | Fully connecting with 64 inputs and 10 outputs |

**Database**   We evaluate our implementation of the homomorphic CNN model on the MNIST dataset to each time calculate ten likelihoods for 32 encrypted images of handwritten digits. The MNIST database includes a training dataset of 60 000 images and a testing dataset of 10 000, each image of which is of size $28 \times 28$. For such an image, each pixel is represented by a 256-level grayscale and each image depicts a digit from zero to nine and is labeled with it.

**Building a model in the clear**   In order to build a homomorphic model, we follow the normal approach for the machine-learning training in the clear — except that we replace the normal `ReLU` function with a polynomial approximation: we (1) train our CNN model described in Table 1 with the MNIST training dataset being normalized into domain $[0, 1]$, and then we (2) implement the well-trained resulting CNN model from step (1) using the HE library and HE programming.

For step (1) we adopt the highly customizable library `keras` with `Tensorflow`, which provides us with a simple framework for defining our own model layers such as the activation layer to enact the polynomial activation function. After many attempts to obtain a decent CNN model, we finally get a CNN model that could reach a precision of $98.66\%$ on the testing dataset. We store the weights of this model into a CSV file for the future use. In step (2) we use the HE programming to implement the CNN model, accessing its weights from the CSV file generated by step (1). We normalize the MNIST training dataset by dividing each pixel by the floating-point constant 255.

**Classifying encrypted inputs**   We implement our homomorphic CNN inference with the library `HEAAN` by [3]. Note that before encrypting the testing dataset of images, we also normalize the MNIST testing dataset by dividing each pixel by the floating-point constant 255, just like the normal procedure on the training dataset in the clear.

**Parameters**. We follow the notation of [10] and set the HE scheme parameters for our implementment: $\Delta = 2^{45}$ and $\Delta_c = 2^{20}$; `slots` $= 32768$; `logQ` $= 1200$ and `logN` $= 16$ to achieve a security level of $80-$bits. (see [8, 9] for more details on these parameters).

**Result**. We evaluate the performance of our implementation on the MNIST testing dataset of 10 000 images. Since in this case `Volley Revolver` encoding method can only deal with 32 MNIST images at one time, we thus partition the 10 000 MNIST testing images into 313 blocks with the last block being padded zeros to make it full. We then test the homomorphic CNN inference on these 313 ciphertexts and finally obtain a classification accuracy of $98.61\%$. The processing of each ciphertext outputs 32 digits with the highest probability of each image, and it takes $\sim 287$ seconds on a cloud server with 40 vCPUs. There is a slight difference in the accuracy between the clear and the encryption, which is due to the fact that the accuracy under the ciphertext is not the same as that under the plaintext. In order to save the modulus, a TensorFlow Lite model could be used to reduce the accuracy in the clear from float 32 to float 16. The data owner only uploads 1 ciphertext ($\sim 19.8$ MB) encrypting these 32 images to the public cloud while the model provider has to send 52 ciphertexts ($\sim 1$ GB) encrypting the weights of the well-trained model to the public cloud.

## 6   Conclusion

The encoding method we proposed in this work, `Volley Revolver`, is particularly tailored for privacy-preserving neural networks. There is a good chance that it can be used to assist the private neural networks training, in which case for the backpropagation algorithm of the fully-connected layer the first matrix $A$ is revolved while the second matrix $B$ is settled to be still.

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

# A   Appendix

Optionally include extra information (complete proofs, additional experiments and plots) in the appendix. This section will often be part of the supplemental material.

