# OpenReview forum: "A Novel Matrix-Encoding Method for Privacy-Preserving Neural Networks (Inference)"
_NeurIPS.cc/2022/Conference — NeurIPS 2022 Submitted_

### Official Review · Reviewer_rzzf · 2022-07-09

**Rating:** 3
**Confidence:** 4
**Soundness:** 2 fair
**Presentation:** 1 poor
**Contribution:** 1 poor

**Summary:**

The paper presents methods for matrix multiplication and 2D convolution in lattice-based homomorphic encryption that allows parallel multiplication of a number of cleartext values. The authors have implemented in an open source framework and provide a benchmark for a simple MNIST network.


**Questions:**

How did you implement SumForConv in terms of rotations and element-wise operations?


**Limitations:**

yes

**Strengths And Weaknesses:**

I'm missing a differentation to the prior work by Dathrati et al. [1]. They propose a matrix multiplication that only uses one ciphertext multiplication as opposed to $2p$ in this work for $p$ being the number of columns in the output, and they report on inference of three versions of a LeNet network with similar accuracy in 2.5 seconds where this work achieves an inference time of 287 seconds for a batch of 32. Even prior art cited by the paper (CryptoNets) achieves 250 seconds on a batch of 4096. The authors claim that memory usage is an issue with CryptoNets but they don't comment on the memory usage of their solution.

I'm also missing the specification SumForConv in terms of basic operations. The underlying encryption scheme only allows element-wise operations and rotations so any other operations has to reduced to these.

Why do the authors encrypt $F$ (line 11 in Algorithm 1 and line 22 in Algorithm 2)? F only contains public information, so a public-private multiplication would be more efficient.

[1] https://dl.acm.org/doi/10.1145/3314221.3314628

---

> ### Author Response · Authors · 2022-08-02
> **Response to reviewer  rzzf**
>
> We would like to thank the reviewers for their input and appreciate their comments.
>
>
> Since $F$ only contains public information, so a constant-ciphertext multiplication would be more efficient. After we completed the  HE programming and paper writing, we did find that it was unnecessary to encrypt the filter matrix $F$. But it was too late to update the source code and paper.
>
> There are three steps to implement SumForConv in terms of rotations and element-wise operations. For example, given a ciphertext $ct_{0}$ encrypting an $h \times w$  image and the kernel size is $kh \times kw$, we could do the following steps: Step 1. we perform a series of the incomplete column shifting on the ciphertext $ct_{0}$, obtaining a new ciphertext $ct_{1}$. Namely, $\texttt{Rot}$($ct_{0}$, $0$)  $\oplus$ $\texttt{Rot}$($ct_{0}$, $1$) $\oplus \cdots $ $\oplus \texttt{Rot}$($ct_{0}$, $kw - 1$) $=$ $ct_{1}$.
> Step 2. we then perform a series of the row shifting on the ciphertext $ct_{1}$, obtaining a new ciphertext $ct_{2}$. Namely, $\texttt{Rot}$($ct_{1}$, $0 \times w$)  $\oplus$ $\texttt{Rot}$($ct_{1}$, $1 \times w$) $\oplus \cdots $ $\oplus \texttt{Rot}$($ct_{1}$, $(kh - 1) \times w$) $=$ $ct_{2}$. Now, the ciphertext $ct_{2}$ has already encrypted the information we want, but with some garbage information. In the final step, we only need to design a special constant vector and perform a public-private multiplication to filter out the garbage information, obtaining the ciphertext we desire.
>
> More comparisons with state-of-the-art work would be made in our future submissions to other conferences or journals. We would like to thank reviewer rzzf for this great suggestion.

---

### Official Review · Reviewer_uaxy · 2022-07-10

**Rating:** 3
**Confidence:** 4
**Soundness:** 2 fair
**Presentation:** 2 fair
**Contribution:** 1 poor

**Summary:**

This paper proposes a new matrix multiplication method (called "volly revolver") suitable for homomorphic encryption. They present an efficient way of adding intermediate steps of the Convolution operation and simulate the operation on the packed ciphertext.

**Questions:**

The paper deals only with matrix multiplication, but it is unclear what advantages there are from multiplication alone. Recent works have already reported a faster computation run-time than reported in the paper. Therefore, a simulation comparison with the latest results is necessary to clarify the advantage of the proposed method in the paper, but there is currently none.

CNN operation does not end with just one matrix operation, but it leads to another operation. In this process, the message size is reduced, and the message packed in the ciphertext needs to be reconstructed for efficient operations. Considering the whole network, convolution operation (and matrix multiplication) should be considered. Otherwise, it can be rather disadvantageous due to additional packing requirements and a corresponding new data rotation for the followed processes.

(215-216 lines) said that "Our method based on Volley Revolver can build convolutional neural networks as deep as it needs." However, they do not

Although this paper targets gray images, other studies handle color images such as CIFAR-10/100 and ImageNet well. Obviously, the proposed method in this paper can also be used for color images, applying three channels independently. However, the amount of computation should be three times larger.

At 78-79 lines, the method in this paper is only applicable to datasets with a smaller number of pixels than the available slots of ciphertext, which is also a limitation of the proposed method. In the case of ImageNet, it is not possible to fit one image even in 2^15 slots, so another efficient method is needed.

On page 4, is the extended form of B included in the single ciphertext? Then, when the extended area is considered, the actually applicable sample size is likely to be further reduced.

It also does not consider how stride two or higher is applicable.

**Limitations:**

The social impact or limitations of the paper are not specifically described.

**Strengths And Weaknesses:**

In applying homomorphic encryption to neural networks, it is necessary to perform matrix operations efficiently. In this process, homomorphic encryption requires an efficient method because the movement of the encrypted message is done through rotation, which has much computational cost.
This paper does not reflect the latest research results. See, for example, C. Juvekar, et al., “GAZELLE: a low latency framework for secure neural network inference,” Proceedings of the 27th USENIX Conference on Security Symposium, August 2018, pp. 1651–1668, which covers how to perform matrix operations for homomorphic operations efficiently.

This paper is not properly compared with homomorphic matrix multiplication results in other papers. There is a limit to directly applying to real CNNs considering only matrix operation without considering the flow of computation in the entire network. The tests conducted only in MNEST appear to be too outdated compared to the recent results of CIFAR-10/100 or ImageNet.

---

> ### Author Response · Authors · 2022-08-02
> **Response to reviewer  uaxy**
>
> We would like to thank the reviewers for their input and appreciate their comments.
>
> Before and after each layer such as the CNN layer, the message size is reduced and the encoding representation is destroyed. Therefore, the message packed in the ciphertext needs to be reconstructed for future use as the input ciphertexts to the next layer.
> Taking the CNN layer in our implementation as an example, after our convolution operation algorithm is finished (our impractical Algorithm 2 running on several simulation virtual ciphertexts), we actually need to do some following work to reconstruct the output ciphertexts to the same encoding representation as the input ciphertexts, which needs about $h$ rotations for an image matrix with $h$ rows. If stride two or higher is adopted, the reconstructing process needs more rotations, about $h \times w$ for an image of size $h \times w$. That is why we don't favor using stride two or higher in our encoding method.  It seems that stride one is the common setting.
>
>
> Volley Revolver can be used for the dataset CIFAR-10/100 under the same parameter setting as in the paper (with the available $2^{15}$ slots of a single ciphertext). Another setting of $logN=17$ and $logQ=1200$ to achieve a $128$-bit security level enables our encoding method to be used to a colored image of size $256 \times 256$, in which case the available slots number is $2^{16}$ and 3 ciphertexts are needed to encrypt one such colored image.  Most common original ImageNet images of size $500 \times 500$ are too harsh to be adopted in the  HE domain and are even unnecessary to be used in the clear. We wonder if there is research work dealing with the original ImageNet dataset.
>
> Future work about building deep CNN in the HE domain via Volley Revolver will be done and the dataset CIFAR-10 will be certainly used in that work.
>
>
> As far as we can see, encrypting the transpose of matrix B for the multiplication A x B is a simple idea but will be widely adopted in future research work.

---

### Official Review · Reviewer_SjJL · 2022-07-11

**Rating:** 3
**Confidence:** 4
**Soundness:** 3 good
**Presentation:** 3 good
**Contribution:** 1 poor

**Summary:**

This work provides the method, named Volley Revolver, of encoding encrypted matrix to perform matrix multiplication efficiently in the HE scheme. Volley Revolver provides an encoding method that packs multiple encrypted data in a single ciphertext, which makes matrix multiplication efficient.

**Questions:**

Is the proposed method applicable for encrypted image data with multiple channels? Most of the colored image dataset such as CIFAR-10 are formed with three channels.

**Limitations:**

The limitation is mentioned in the introduction (lines 40--41).

**Strengths And Weaknesses:**

Strengths:

The authors described the details of the proposed method with proper figures which are easy to understand.

---

Weaknesses:

The authors use the word *efficient* a lot, but its meaning should be described more definitely. From my understanding, the meaning of "efficient" refers that multiple image data can be encoded in a single ciphertext. If this is the authors' main contribution, please emphasize this point in Abstract, Introduction, Related Works, and Conclusions.

Moreover, there is no quantitative discussion about the efficiency of the proposed method. Previous works are mentioned, but I cannot find the exact improvement compared to the previous works. Are there any other previous works about matrix encoding for matrix multiplication or convolution operation? If so, the authors should refer those works. If not, please mention it. And please show the performance which indicates the improvement from the previous works (e.g., runtime, the number of multiplications). Without these discussions, I think the proposed method is hard to be considered as an efficient method.

---

> ### Author Response · Authors · 2022-08-02
> **Response to reviewer  SjJL**
>
> We would like to thank the reviewers for their input and appreciate their comments.
>
> We use the word efficient a lot. Here we mean that the main frameworks of the matrix computation and convolution operation can be computed in parallel.  Multiple image data being encoded in a single ciphertext allows one ciphertext to encrypt semantically-complete information for each image, facilitating the design of parallel algorithms. The response time varies significantly on how many vCPUs the cloud has. For example, we first test our final artifact on a server with 12 vCPUs, and the response is 30 minutes or so for 32 MNIST images. The experimental result in our paper is obtained by running our CNN implementation on a cloud with 40 vCPUs, taking about 285 seconds to respond.
>
> Our encoding method is applicable for encrypted image data with multiple channels. Suppose that there are 32 32x32 color images (CIFAR-10 images) with three channels. For simplicity, we adopt the same parameter setting as in the paper. For this toy example, our encoding method only needs to use 3 ciphertexts to encrypt the three channels of the 32 color images respectively. Each ciphertext encrypts the corresponding channel of the 32 colored images just like our encoding method encrypting the MNIST grey images. Note that in this special case, not a single slot of the three ciphertexts is wasted, which is the best optimum ciphertext size.    In conclusion, our encoding method needs 3 ciphertexts to encrypt one colored image.

---

> > ### Comment · Reviewer_SjJL · 2022-08-08
> > **Response for rebuttal**
> >
> > I agree that the proposed method provides parallel algorithms which make the inference more efficient. However, to be approved of the novelty of the efficiency, as I mentioned previously, a comparison with the previous method is necessary. In my opinion, if the quantitative comparison with other previous work is hard to be done at this moment, the paper should be submitted later after the comparison is done. (e.g. runtime.) The authors should analyze or set a fair experiment to compare with other previous works.

---

> > > ### Author Response · Authors · 2022-08-08
> > > **Response to reviewer SjJL**
> > >
> > > I have noticed that this paper lacks a comparison with previous methods.
> > >
> > > And I am very grateful for the time you and other reviewers spent reading my work.
> > >
> > > I hope this work doesn't waste your time.

---

### Official Review · Reviewer_sp2h · 2022-07-11

**Rating:** 2
**Confidence:** 4
**Soundness:** 1 poor
**Presentation:** 3 good
**Contribution:** 1 poor

**Summary:**

This work deals with the new matrix encoding method, called Volley Revolver, for efficient matrix multiplication on homomorphic encryption scheme. To efficiently use the data structure and the homomorphic operation in CKKS scheme for matrix multiplication, they adequately transpose and concatenate one of the input matrices and use RowShifter and SumColVec operation. The matrix multiplication and convolution operation are efficiently designed for CKKS scheme, and the simulation is conducted with HEAAN library and MNIST dataset.

**Questions:**

- Why Jiang et al.'s work is difficult to be applied in CNN with over two convolutional layers?
- Please show the comparison with the state-of-the-art prior techniques.

**Limitations:**

They did not deal with the limitation of this work, and I think there is no potential negative societal impact.

**Strengths And Weaknesses:**

Strengths
- The authors deal with matrix multiplication and convolution with the general dimensions of matrices.
- The ideas are well conveyed with specific matrix formulas to ease the understanding.

Weaknesses
- No simulation comparison with state-of-the-art matrix multiplication or convolution with CKKS scheme. As far as I know, the matrix multiplication and convolution in CKKS scheme is well researched in homomorphic encryption academic area, and thus this proposed technique can be compared with many prior works regarding matrix multiplication. They only show the classification accuracy and the latency with their proposed model without any comparison, so I cannot be assured that the proposed technique is superior to the state-of-the-art techniques.
- Reference is too weak. There are far more works regarding matrix multiplication and convolution in HE, but they only refer to only a small subset of these works.
- They refers to the work of Jiang et al. [9] and they claimed that the technique in this work "might" be difficult to be applied in practical application for CNN with over two convolutional layers. However, the authors do not suggest why it is difficult. The authors should have dealt with the limitation of the prior works thoroughly before proposing their technique. I cannot be assured that the prior techniques are really inadequate for CNN with over two convolutional layers.
- Overall, the authors failed to prove that their proposed technique is superior to the state-of-the-art techniques, and thus the authors should justify their contribution more strongly if they want to be accepted in the NeurIPS conference.

---

> ### Author Response · Authors · 2022-08-02
> **Response to reviewer sp2h**
>
> We would like to thank the reviewers for their input and appreciate their comments.
>
> From our own perspective, the main reason that Jiang et al.'s work is difficult to be applied in CNN with over two convolutional layers is that the encoding representations before the CNN layer and after the CNN layer are too different. Supposing that there is another CNN layer stacking after the first CNN layer, it would take a lot of HE operations to reconstruct the encoding representation of the output ciphertexts of the first CNN layer to the same as that of the input ciphertexts of the following CNN layer. That is why we think it is difficult for us to use the baseline method to implement CNN with over two CNN layers. However, Jiang et al. know their method better than we do and probably would come up with another novelty idea to overcome the problem to us.
>
> The only advantage of our method in the Neural Networks Inference compared to the state-of-the-art techniques is probably that our encoding method only takes one ciphertext to encrypt several images while others do not. Other works need several ciphertexts to encrypt several images. More comparisons with state-of-the-art work would be made in our future submissions to other conferences or journals. We would like to thank reviewer sp2h for this great suggestion.

---

> > ### Comment · Reviewer_sp2h · 2022-08-07
> > **Response to the comment of the authors**
> >
> > I see the point of the authors. However, the main problem of this work is the lack of sufficient investigations for prior works. The main contribution of Jiang et al.'s work is the general matrix multiplication on the homomorphic encryption scheme, not the convolution operation itself. There are so many prior works for the convolution operation on the homomorphic encryption schemes. For example, did you see the following paper?
> >
> > Juvekar, Chiraag, Vinod Vaikuntanathan, and Anantha Chandrakasan. "GAZELLE: A low latency framework for secure neural network inference." 27th USENIX Security Symposium (USENIX Security 18). 2018.
> >
> > This is not even the recent work about the convolution operation on homomorphic encryption, and there have been many prior works based on this work. See the papers that referred to Gazelle. The authors have to consider these works.

---

> > > ### Author Response · Authors · 2022-08-08
> > > **Response to reviewer sp2h**
> > >
> > > I don't think I have read the paper on GAZELLE or the one about MiniONN (both look familiar to me, though).
> > >
> > > I just read four or five papers based on only HE technique and came up with the basic ideas behind this work. Jiang et al.'s work gave me a lot of inspiration and thinking.
> > >
> > > Indeed, I did few investigations for prior works.
> > >
> > > I will see the papers that referred to Gazelle if I can restart my Ph.D. study again. I hope so.

---

### Meta-Review · Area_Chair_1eRY · 2022-08-25

**Recommendation:** Reject
**Confidence:** Certain

**Metareview:**

The reviewers were unanimous in their recommendation to reject the paper. The authors' responded to the reviews but recognized the limitation of their submission, particularly in terms of missing comparisons to related work.

I want to take this opportunity to address the author who wrote in their rebuttal:

 *"I will see the papers that referred to Gazelle if I can restart my Ph.D. study again. I hope so."*

I sincerely hope you have the chance to restart your Ph.D. program and continue your research. The conference review process can be daunting -- yet it is an important step in pushing our field forward. In the case of your paper, the reviewers appreciated your ideas and the quality of the presentation, describing it as clear and easy to understand. What was missing was a more comprehensive comparison with related work -- a common misstep, even for seasoned researchers. Please **do not** let this discourage you from engaging in research. In fact, I hope this experience demonstrates the value of peer review, serves as a learning experience, and helps your write a better paper.

I look forward to crossing paths with your work again.


**Award:**

No

---

### Decision · Program_Chairs · 2022-09-14

Reject